# Is Africa Ready to Use Phycoremediation to Treat Domestic Wastewater as an Alternative Natural Base Solution? A Case Study

**Paul J. Oberholster [1],\*, Yolandi Schoeman [1] and Anna-Maria Botha [2]**

1   Centre for Environmental Management, University of the Free State, Bloemfontein 9300, South Africa; schoeman.y@ufs.ac.za
2   Department of Genetics, University of Stellenbosch, Private Bag X1, Matieland, Stellenbosch 7601, South Africa; ambo@sun.ac.za
\*   Correspondence: oberholsterpj@ufs.ac.za; Tel.: +27-(0)514013740

**Abstract:** This review outlines the potential of phycoremediation as a natural, cost-effective solution for domestic wastewater treatment in Africa, particularly focusing on its application in less densely populated and rural areas. The urgency of improving sanitation access, a key objective in both the Millennium Development Goals (2000–2015) and the Sustainable Development Goals (2015–2030), is underscored by the fact that half of Africa's population suffers from diseases linked to inadequate water and sanitation facilities. South Africa, a focal point of this study, faces significant challenges in wastewater management. These include the limited capacity of wastewater treatment plants to handle the burgeoning wastewater volumes due to population growth, unregulated discharges causing fluctuating pollution levels, and high operational costs leading to improper sludge disposal and odor issues. Compounding these problems are frequent power outages, financial constraints impacting wastewater treatment plant operations and maintenance across Africa, and a lack of skilled personnel to manage these facilities.

**Keywords:** phytoremediation; bioreactors; algal cultures; wastewater treatment; energy efficiency; optimized phycoremediation algal pond system (OPAPS)

## 1. Introduction

Waterways in South Africa are currently burdened with preventable pollutants, a situation exacerbated by institutional limitations, aging infrastructure, and increased hydraulic loads. Africa is grappling with a complex water crisis, characterized by a myriad of challenges. These include ongoing droughts, a disparity between water supply and demand, deteriorating infrastructure, leaks and water losses through theft and vandalism (see Figure 1), as well as a loss of essential skills. Contributing factors also encompass an underfunded and ineffective education system, management failures, and a decline in water quality, all of which pose significant threats and concerns [1].

The escalating demands for water amid a growing global population and the impending challenges of climate change necessitate a paradigm shift in wastewater management. The current crisis is primarily rooted in outdated infrastructure and inadequate systems for handling the surge in wastewater volumes [2]. This situation is particularly acute in Africa, where the reuse of contaminated wastewater and limited access to sanitation are leading to increased incidences of disease and illness [3]. Presently, half of the African population is afflicted by diseases linked to the insufficient provision of water and sanitation. The continent faces alarming health statistics, such as the death of 155 children every hour from water, hygiene, and sanitation-related diseases, and it records the highest rates of cholera and diarrhea among children [4]. A comprehensive study by Hickling and Hutton [5],

covering 18 African countries, encompassing approximately half of the continent's population, revealed that premature deaths directly or indirectly associated with sanitation issues account for 48−90% of total economic costs in these nations. In Burkina Faso, for example, the annual economic burden of poor sanitation was estimated at USD 136 million, with 88% of diarrheal deaths attributed to fecal−oral transmission. Kenya alone incurred USD 2.7 million per year in lost productivity due to time off from work or school, seeking medical treatment, and caring for young children, all attributable to inadequate sanitation [5].

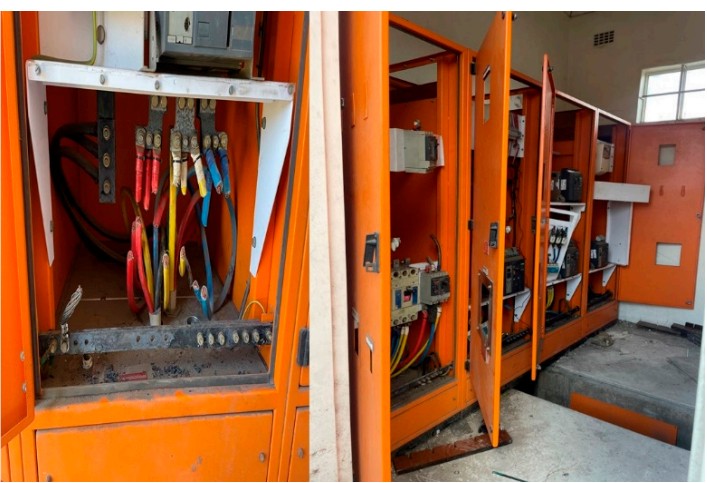

**Figure 1.** Theft and vandalism at a conventional wastewater treatment plant (WWTP) in Mpumalange province, South Africa (courtesy of KD Mabitsela).

As Table 1 illustrates, among the technical, social, economic, and environmental challenges, a critical issue is the insufficient capacity of WWTPs to handle the increasing wastewater volumes driven by population growth. Compounding this issue is the need to manage fluctuating pollution loads resulting from unregulated discharges, such as industrial waste, into the sewage network. Additionally, the financial burden of high operational and maintenance costs often leads to suboptimal sludge disposal practices and odor problems. Power outages further exacerbate these challenges, particularly in scenarios where treatment processes are energy-dependent.

Financial constraints are a recurring problem across the countries studied, adversely impacting the operation, maintenance, and upgrading of WWTPs [1]. The release of inadequately treated wastewater into the environment, a common occurrence in countries like Ghana, is often a result of dysfunctional or temporarily disconnected treatment plants. Moreover, the workers and managers responsible for operating these plants often lack the necessary skills and motivation to maintain them effectively. This can be partly attributed to inadequate compensation and training [1].

In South Africa, water supply and sanitation present a complex picture of both significant achievements and persistent challenges. The nation boasts a robust water industry known for innovation, yet progress in sanitation remains limited. Access to sanitation is crucial, not only for human dignity but also for its direct correlation with disease control and potential impacts on water resources. Between 2006 and 2011, access to sanitation services in South Africa improved notably, with households that gained access to flush toilets connected to a public sewerage system increasing from 51.9% in 2001 to 60.1% in 2011 [6,7].

**Table 1.** Summary of reported challenges at wastewater treatment plants (WWTPs) in the selected countries (adopted from Nikiema et al. [1]).

| Parameters | | Algeria | Burkina-Faso | Egypt | Ghana | Morocco | Senegal | South Africa | Tunisia |
|---|---|---|---|---|---|---|---|---|---|
| | Technical | Power cuts Industrial waste-water inputs (e.g., presence of oil) Sludge discharge | No control over industrial disposal Power cuts Limited removal of nitrate or iron Lack of compliance with the regulations | High loading rates Lack of spare parts Limited infrastructure for biogas reuse | Pump failure Power cuts Overloading | Pump failure Power cuts Lack of control over wastewater feed Foaming in activated sludge WWTPs Poor management of sludge production | No control over industrial disposals Power cuts Limited removal of nitrate or iron Lack of compliance with the regulations | Power cuts High loading rates Poor maintenance | Sludge elimination |
| | Social | Need of capacity building for sludge management | Solid waste disposed in the collection network Robbery Vandalism | Need for capacity building for sludge management Low wages of workers causing lack of motivation | Waste thrown in sludge Complaints about odor and breeding of mosquitoes | Limited qualified personnel Inadequate standards and regulations | Pump failure Power cuts Overloading | Limited qualified personnel Robbery Vandalism | |
| | Economic | Outdated equipment | High operational and maintenance costs | High operational and maintenance costs High cost of WWTPs | Lack of funds for operational and maintenance costs or rehabilitation High operational and maintenance costs | Inadequate infrastructure High operational and maintenance costs | Non-sustainable funding sources Lack of funds for operational and maintenance costs (e.g., fuel for generator) | Poor governance and misappropriation of funding Lack of funds for operational and maintenance costs (e.g., fuel for generator) | High energy consumption |
| | Environmental | | | Water reuse should be optimized at least for forest trees | Odor affects local communities in the vicinity of the WWTP | Air pollution (e.g., release of odors) | Deterioration of living conditions of populations Groundwater pollution Ecosystem disturbance | Deterioration of living conditions of populations Groundwater pollution Ecosystem disturbance | |

Moving forward, the governance of water resources becomes pivotal for sustainable human development, economic growth, and poverty alleviation. This will require proactive investment in wastewater treatment infrastructure, the adoption of innovative policy approaches, and exploring alternative funding mechanisms [8]. Reconceptualizing wastewater, not as a problem but as a potential resource, is crucial. Effectively managed wastewater can be an invaluable asset, enhancing food security, health, and economic well-being globally. However, poor management can pose a significant risk due to the pollution of aquatic environments from untreated sewage, for example, phosphorus concentrations exceeding 1 mg/L in water bodies, posing a serious threat to environmental, animal, and aquatic life [9]. Phosphorus is a critical nutrient that exacerbates algal blooms, leading to eutrophication. This process depletes oxygen in water bodies due to algal decay, adversely affecting aquatic life. The discharge of sewage and other pollutants downstream of urban areas intensifies eutrophication, leading to toxic cyanobacterial blooms that can be detrimental to public health [10]. The increasing dominance of these toxic blooms in eutrophic lakes is a growing concern for water utility managers worldwide [11].

In response, this paper highlights the optimization of phycoremediation, a process offering multiple benefits. Phycoremediation is a low-cost, electricity-free, and chemical-free approach that utilizes existing infrastructure. It is environmentally friendly and straightforward to implement and operate, making it particularly suitable for enhancing treatment capacity in small, rural wastewater treatment plants (WWTPs). This review will mostly focus on the direct use of microagal species due to their high potential to convert solar to chemical energy [12,13], and posit that optimizing phycoremediation presents a viable, eco-sensitive, and cost-effective medium-term solution for these areas. It emphasizes the feasibility of this method in terms of resource utilization and operational simplicity, potentially revolutionizing wastewater treatment in rural Africa.

## 2. Phycoremediation as a Potential Solution

Phycoremediation, as defined by Liu et al. [14], involves the use of micro- or macro-algae to remove or biotransform pollutants, including nutrients and toxic chemicals, from various types of wastewater. It seems to provide a sustainable solution by decreasing the carbon footprint (atmospheric carbon dioxide that causes pollution and global greenhouse effects) since algae serves as a good sink for carbon dioxide. Additionally, the proportionality between efficiency and algal growth, also implies that factors that promote microalgal development will enhance algal biomass [13,15]. This method has been applied in municipal wastewater treatment for more than six decades, with its first documented use reported by Oswald [16]. A significant body of research focuses on utilizing microalgae to extract nitrogen and phosphates from domestic effluents, thereby mitigating eutrophication [17]. Recently, the application of phycoremediation using microalgae for nutrient removal from wastewater has garnered increasing interest [18].

Phycoremediation encompasses the treatment of pollutants in contaminated areas using both micro and macroalgae [19]. Microalgae can be effectively cultured for treatment purposes in various water types, including fresh, marine, and brackish waters. During their photosynthetic process, microalgae use atmospheric carbon dioxide and have demonstrated significant potential for greenhouse gas abatement. Chisti [20] noted that algae reproduce rapidly, exhibiting faster growth rates than any energy crop, and can be harvested frequently. The applications of microalgae extend beyond pollution abatement to include biofuel production and carbon sequestration [21].

Phycoremediation stands out as an eco-friendly, low-cost technology, presenting a particularly attractive solution for pollution control in developing countries [22]. The presented case study demonstrates the feasibility of optimizing phycoremediation as a method for treating domestic wastewater and reducing nutrient levels in African waterbodies and identifies potential challenges and lessons learned from a case study conducted under South African environmental conditions.

## 3. Waste Stabilization Ponds

Waste stabilization ponds (WSPs) are among the most common and effective wastewater treatment methods globally (Table 2). According to Tilley et al. [23], WSPs are defined as "large, man-made water bodies where blackwater, greywater, or faecal sludge are treated through natural processes, influenced by solar light, wind, microorganisms, and algae". These ponds are particularly suitable for rural communities with access to expansive open lands, distanced from residential and public areas, where developing a local collection system is feasible. The efficiency of WSPs is significantly influenced by the intensity of sunlight and ambient temperature, making them particularly effective in tropical and subtropical regions [24]. Table 2 presents a summary of the main constituents of wastewater and stormwater and the advantages and disadvantages of WSPs.

**Table 2.** Main constituents in wastewater and stormwater and the advantages and disadvantages of waste stabilization ponds [24] (adopted from Varón and Mara [24]).

| Constituents | Representative Parameters | Source/Relevance | | | Possible Effects of the Hazard |
|---|---|---|---|---|---|
| | | Wastewater | | Urban Stormwater | |
| | | Domestic | Industrial | | |
| **Pathogens** | *E. coli* Coliforms | High | Variable | Medium | Waterborne diseases |
| **Suspended solids** | Total suspended solids | High | Variable | Medium | Sludge deposits Hazard adsorption Shielding of pathogens against disinfectants; affecting treatment |
| **Bio-degradable organic matter** | Biochemical oxygen demand | High | Variable | Medium | Oxygen consumption Death of fish Septic conditions |
| **Nutrients** | Nitrogen Phosphorus | High | Variable | Medium | Excessive growth of cyanobacteria and algae Toxicity to fish (ammonia) Oxygen consumption Illnesses in new-born infants (nitrate) Pollution of groundwater (nitrate) |
| **Poorly biodegradable organic matter** | Some pesticides Some detergents Pharmaceuticals | Medium | Variable | Low | Toxicity (various) Foam (detergents) Reduction of oxygen transfer (detergents) Reduced or non-biodegradability Offensive odors (e.g., phenols) |
| **Heavy metals** | Specific elements (e.g., arsenic, cadmium, chromium, copper, lead, mercury, nickel, and zinc) | Medium | Variable | Low | Inhibition of biological sewage treatment Contamination of groundwater |
| **Inorganic dissolved solids** | Total dissolved solids Conductivity | Medium | Variable | Not relevant | Excessive salinity—harm to plantations (irrigation) Toxicity to plants (some ions) Problems with soil permeability (sodium) |

WSPs consist of several ponds that are categorized into three types based on their operational characteristics and biological processes: (1) anaerobic, (2) facultative, and (3) aerobic (maturation) ponds. Anaerobic ponds function in the absence of dissolved oxygen and are designed to handle high organic loads. The biochemical oxygen demand (BOD) reduction in these ponds is primarily achieved through the sedimentation of solids and subsequent anaerobic digestion within the formed sludge layer. A typical retention time in anaerobic ponds ranges from one to one and a half days [25]. Facultative ponds operate under aerobic conditions at the surface and anaerobic conditions at the bottom sediment layer [26]. These ponds are further divided into primary and secondary facultative ponds. Primary facultative ponds receive raw wastewater, while secondary ones treat particle-free effluent. The design of facultative ponds focuses on BOD removal, facilitated by a healthy algal population that generates oxygen through photosynthesis for bacterial activity. The bottom layer of primary facultative ponds contains sludge deposits decomposed by anaerobic bacteria [25], while aerobic (maturation) ponds, also known as polishing ponds,

receive effluent from secondary facultative ponds. Maturation ponds are characterized by minimal vertical stratification and maintain good oxygenation throughout the day. They are primarily designed for pathogen removal, with their size and number determined by the required bacterial quality of the final effluent [23,25]. Algal diversity in maturation ponds is typically higher than in facultative ponds, with non-motile genera being more prevalent. Algae play a crucial role in these ponds, absorbing phosphates, carbon dioxide, and nitrogen compounds, and simultaneously providing oxygen for heterotrophic bacteria to decompose organic material. Tilley et al. [23] noted that when used in conjunction with algae and/or fish harvesting, maturation ponds can effectively remove most nitrogen and phosphorus from the effluent, such as in the case of the Motetema wastewater treatment pond system.

## 4. Case Study Area Background

The Sekhukhune district where the WWTP is located, spans approximately 13,264 km$^2$, and is home to a population of 1,122,522. Predominantly rural, the district comprises nearly 740 villages, with a sparse average population density of 83.0 persons per square kilometer, dispersed across the area [27]. Only about 5% of the population resides in urban settings. Economically, the district is categorized in socioeconomic quintile 1, marking it as one of the poorest in the region [28]. Statistics South Africa's census figures indicate that the Greater Sekhukhune district had the highest unemployment rate (50.9%) in the Limpopo province [7]. The greater Sekhukhune District Municipality faces significant challenges in water quality and sanitation services. The Green Drop Report [29] identified a regressive trend in 13 of the 16 WWTPs assessed in the area, with three plants at high risk and 13 at critical risk. The effluent from these plants pollutes local water bodies, undermining ecosystem services and posing substantial health risks to downstream communities.

Given the lack of advanced WWTP infrastructure and electricity, the Motetema WWTP utilizes pond systems to treat domestic waste for a population of about 11,400. Discussions with the Sekhukhune District Municipality revealed that the increasing burden on the municipality is due not only to the growing population but also to general governance issues. These include aging infrastructure, inadequate technical skills, and limited financial resources. There is a consensus on the urgent need for sustainable, long-term solutions. Key risk factors identified include inadequate effluent monitoring, non-compliant effluent quality, insufficient flow monitoring, and non-compliance with Regulation 813 regarding technical skills requirements. The district's regression in 13 of its 16 WWTPs highlights the high environmental and health risks currently posed by existing practices [29].

### 4.1. Motetema Wastewater Treatment Pond System

The Motetema WWTP consists of two sets of six operating ponds and serves roughly 1560 households (Figure 2). The discharged effluent frequently fails to meet national and provincial regulations, posing significant risks to the environment, the natural water sources, and the health of humans and animals. In rural areas in South Africa, treatment ponds like these are common for decentralized domestic sewage treatment. They are cost-effective, relying primarily on natural processes without external energy inputs. Employing algae to assimilate nutrients before discharging wastewater into phosphate-sensitive rivers offers an environmentally friendly, cost-effective solution.

The Motetema WWTP operates without mechanical aeration and is designed to treat an average total effluent of 2.5 ML per day. However, in practice, the facility receives approximately 4.5 ML/day, nearly double its intended capacity (see Table 3). The system comprises 12 ponds, organized into two series of six ponds each. At any given time, only one series of six ponds is operational, while the other set undergoes cleaning. This pond system functions on a gravity-based overflow mechanism, transferring effluent from one pond to the next.

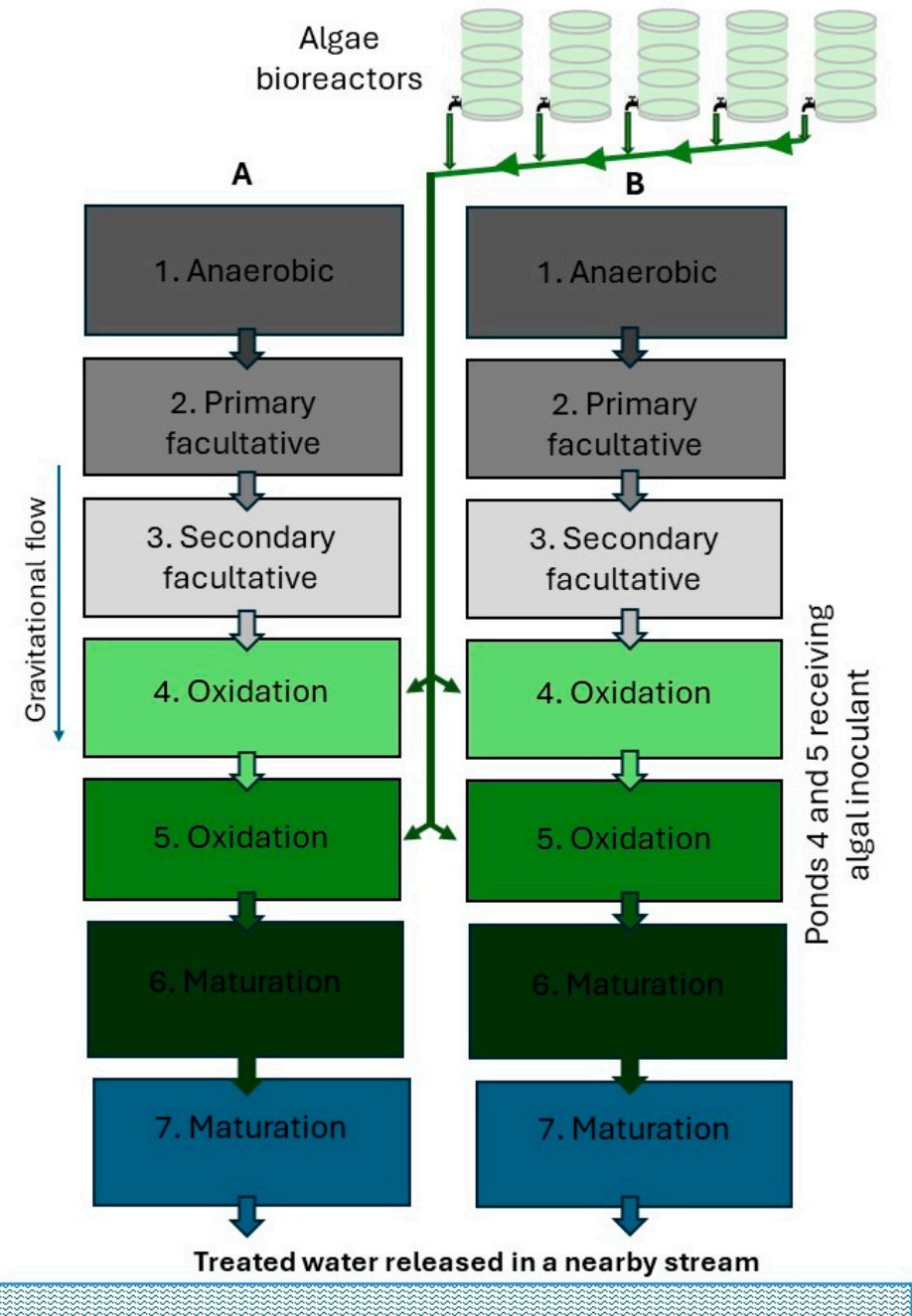

**Figure 2.** Layout of the Motetema water treatment pond system.

**Table 3.** Dimensions and categories of the seven different ponds at the Motetema WWTP.

| Pond | Depth (m) | Area (m²) | Volume (m³) | Category |
|---|---|---|---|---|
| 1 | 2.5 | 38,571.43 | 96,428.57 | Anaerobic |
| 2 | 2 | 9183.67 | 18,367.35 | Facultative |
| 3 | 2 | 5969.39 | 11,938.78 | Aerobic/maturation |
| 4 | 1.5 | 4336.73 | 6505.10 | Aerobic/maturation |
| 5 | 1.5 | 4132.65 | 6198.98 | Aerobic/maturation |
| 6 | 1.5 | 10,204.08 | 15,306.12 | Aerobic/maturation |
| 7 | 2 | 16,836.73 | 25,255.10 | Aerobic/maturation |

As with most wastewater pond systems, the Motetema WWTP utilizes a series of interconnected ponds for enhanced wastewater treatment, namely anaerobic, facultative, and aerobic (maturation) ponds.

The efficient functioning of any natural-based treatment system, such as in the case of the Motetema WWTP, is not only dependent on political or socioeconomic factors such as funding, infrastructure, maintenance, and management; the key to success is also underpinned by biological processes. Hence, in the case of Motetema, we followed a two-phase approach. The first phase of the process was to identify microalgal species that would be able to grow, proliferate, and treat the wastewater. Once the microalgal species were selected based on initial experimentation, in the second phase, we upscaled our treatment system to optimize the treatment at Motetema WWTP. During this phase, the algal photobioreactors were implemented (Figure 3) and used for pond treatment, and the outflow was assessed for treatment capacity.

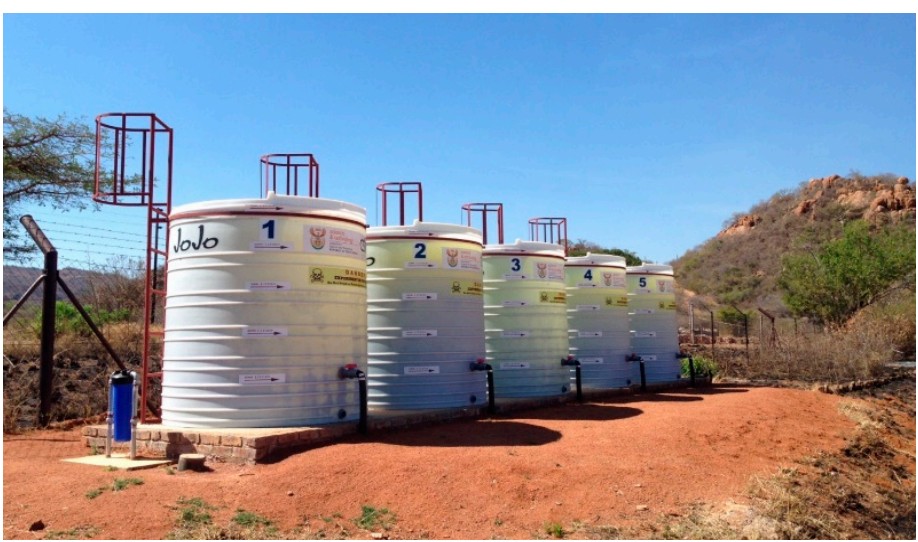

**Figure 3.** Photobioreactors at the Motetema wastewater treatment pond system implemented during Phase 2 [30].

### 4.1.1. Phase 1: Selection of Suitable Microalgal Species

Two microalgal species from the Chlorophyte phylum, specifically *Chlorella vulgaris* and *Chlorella protothecoides*, were selected for laboratory experimentation based on several criteria: (1) their high phosphate uptake potential; (2) rapid exponential growth; and (3) wide temperature tolerance range [12,13,31]. These species were previously cultured under various environmental conditions in the laboratory. The microalgae, either *C. vulgaris*, *C. protothecoides*, or a combination of both, were exposed to wastewater effluent at a ratio of 1:1000 based on total cell counts. A control sample consisted of wastewater effluent without the selected algae. The purpose was to observe the algae's competition with indigenous bacteria and algae species. The experiments were conducted in a horizontal laminar flow cabinet, with regular microscopic inspections to test inoculants for contamination (Supplementary File S1). The growth rates of the total algal biomass are expressed in terms of total chlorophyll, while long-term axenic laboratory cultures of *Chlorella* were maintained through a routine serial subculture over a three-month period.

The growth patterns, in terms of cell counts and chlorophyll content, for the three different algal exposures under laboratory conditions are illustrated in Figure 5. An initial lag phase was observed in all four curves during the first three days. This was followed by a logarithmic growth phase over the next four days for *C. protothecoides* and the combination of *C. vulgaris* and *C. protothecoides*. In contrast, the control group (Motetema domestic wastewater) and samples with only *C. vulgaris* showed no significant growth increase. Notably, the combined algae culture demonstrated better proliferation compared

to *C. vulgaris* alone when exposed to Motetema domestic wastewater ($p < 0.05$). This pattern was similar to that observed for *C. protothecoides*, suggesting that both the combination of *C. vulgaris* and *C. protothecoides*, and the single culture of *C. protothecoides* thrive in the Motetema domestic wastewater. Pigment analysis, specifically for chlorophyll *a* and *b*, correlated with the data on total cell counts (Figure 4).

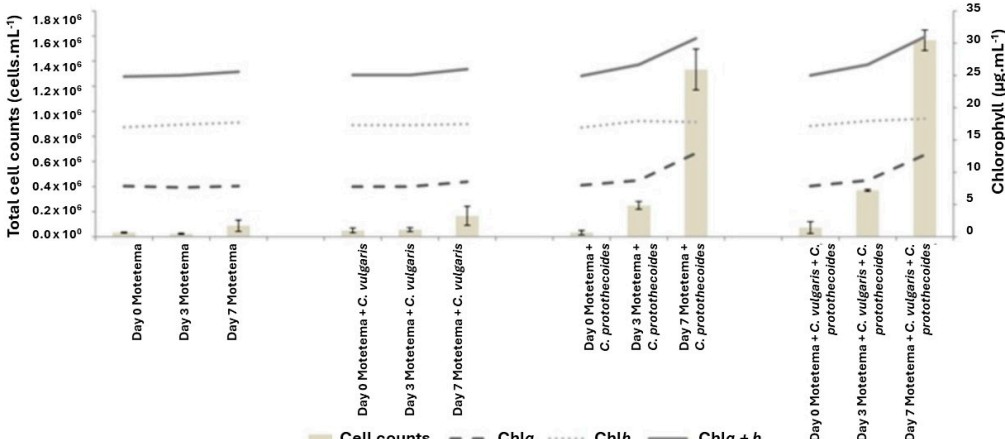

**Figure 4.** Growth patterns in terms of total cell counts and total chlorophyll content for the three different algal exposures (i.e., *C. vulgaris, C. protothecoides,* and the combination of *C. vulgaris* and *C. protothecoides*) over a period of seven days under laboratory conditions. The control consisted of only the unfiltered domestic wastewater from Pond 4 at the Motetema WWTP (Courtesy of P-H Cheng).

Chlorophylls *a* and *b* were measured, as these pigments are vital components for capturing light energy for photosynthesis. Both contain a central magnesium ion encased in a porphyrin ring structure. While chlorophyll *a* is crucial for the release of chemical energy, it is not the sole pigment involved in photosynthesis. Chlorophyll *b*, more soluble in polar solvents than Chlorophyll *a*, is closely associated with photosystem II. It plays a significant role under low light intensity by increasing the ratio of photosystem II to photosystem I.

The laboratory study, focusing on the exposure of *C. vulgaris* and *C. protothecoides* to Motetema domestic wastewater, yielded noteworthy findings. The obtained data indicated that both *C. vulgaris* and *C. protothecoides* were effective in outcompeting indigenous algae present in Pond 4. However, when exposed to Motetema domestic wastewater, *C. vulgaris* exhibited less significant growth compared to a combination of *C. protothecoides* and *C. vulgaris*. Hence, Phase 2 of the process was initiated using the combination of microalga *C. protothecoides* and *C. vulgaris* as targets for testing the treatment process.

### 4.1.2. Phase 2: Mass Inoculation for Optimization of Phycoremediation

In the Motetema wastewater treatment pond system, *C. vulgaris* and *C. protothecoides* were selected for mass culturing and subsequent inoculation. Specifically, maturation Ponds 4 and 5 were chosen for this inoculation process (Figure 5). The inoculation of these ponds with the selected consortium of algae was carried out every four to five weeks, with the frequency adjusted according to seasonal variations (summer and winter). The concentration of the algal consortium introduced into the ponds was maintained at 10,000 cells.mL$^{-1}$.

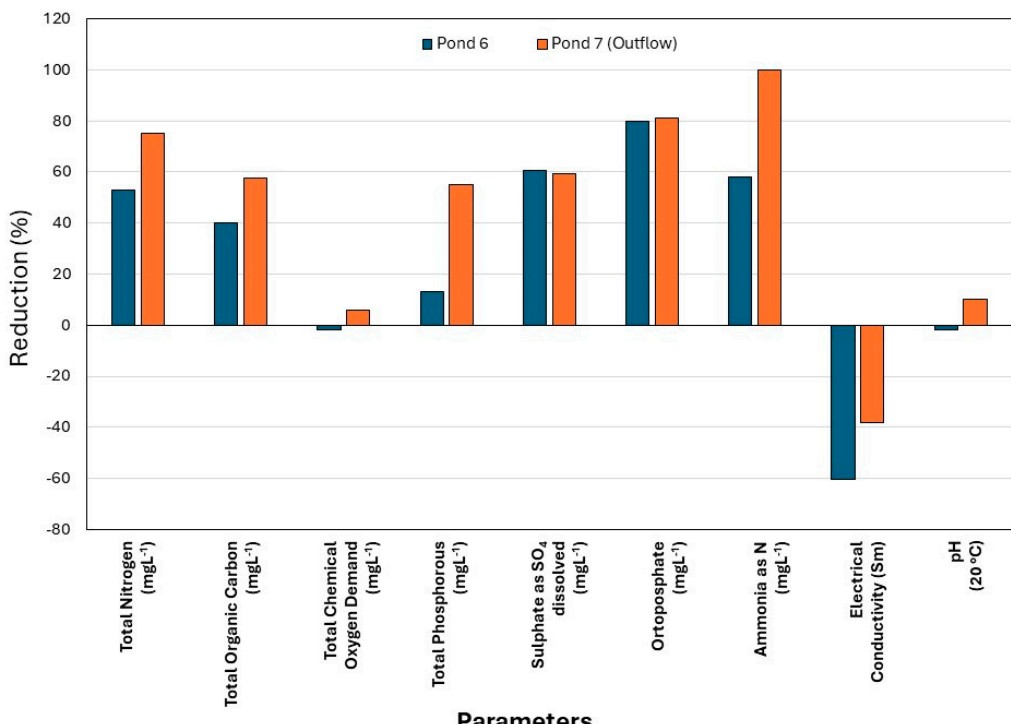

**Figure 5.** Reduction of selected parameters one year after treatment with *C. vulgaris* and *C. protothecoides* at the Motetema wastewater treatment pond system.

Given the design of the pond system, which relies on natural overflow for water movement from one pond to another, it was anticipated that the inoculated algae would naturally disseminate from the inoculated ponds to adjacent ones. This expectation was based on the assumption that the algae, once introduced into Ponds 4 and 5, would be carried along by the water flow to the subsequent ponds in the system.

For the evaluation of physical, chemical, and biological characteristics in the Motetema WWTP, samples were consistently collected from the outlets of Ponds 6 and 7. Two distinct time points were chosen for sample collection, namely prior to algae inoculation and one year after the commencement of continuous inoculation, while standard in situ measurements were taken, namely dissolved oxygen (DO), temperature (°C), pH, and electrical conductivity (EC).

To assess the effectiveness of phycoremediation through the mass inoculation of microalgae, water samples from the last two maturation ponds (6 and 7) of the Motetema WWTP were analyzed before and after treatment over a period of one year. The initial samples were collected prior to any microalgae inoculation, while the final samples were taken four weeks after the last of a series of mass microalgae inoculations in these ponds over the course of one year.

A comparative analysis of the water quality data from the final effluent of Pond 7, before and after one year of algae treatment, revealed significant reductions in key pollutants. We found that orthophosphate levels decreased from 8 mgL$^{-1}$ to 1.36 mgL$^{-1}$, achieving an 83% reduction, while ammonia levels showed a 99% reduction (from 19 mgL$^{-1}$ to 0.1 mgL$^{-1}$), and total nitrogen was reduced from 41 mgL$^{-1}$ to 11 mgL$^{-1}$, marking a 73% reduction (Figure 5). These substantial reductions in nutrient levels indicate a decreased likelihood of eutrophication in water bodies receiving this effluent. According to the literature [32] the rate of gaseous NH3 loss to the atmosphere is primarily a function of a high pH, the surface to volume ratio of the maturation pond, temperature, and the mixing conditions in the maturation pond. A high alkaline water pH shifts the equilibrium of $NH_3$ gas and $NH_4$ towards gaseous $NH_3$ production, while the mixing conditions in the maturation pond affect the magnitude of the mass transfer coefficient.

Microscopic analysis confirmed that both *C. vulgaris* and *C. protothecoides* became the dominant algae species in Ponds 6 and 7 after one year of consistent inoculation [31]. This suggests that the targeted mass inoculation strategy was effective in enhancing the phycoremediation capacity of these wastewater treatment ponds.

Despite the significant improvement in the quality of the wastewater after treatment in the pond system, several challenges were experienced during the first year of mass inoculation of selected algae at the Motetema WWTP. These challenges are summarized in Table 4.

**Table 4.** Challenges experienced during Phase 2 during the optimization of the phycoremediation at the Motetema WWTP.

| Experienced Challenge | Reason for Prevailing Challenge | Impact of Experienced Problem |
|---|---|---|
| Duckweed overgrowth | Overgrowth in the last maturation pond reduced light penetration, affecting algae photosynthesis | Hindered phycoremediation due to reduced algae photosynthesis |
| Field fires | Frequent field fires, related to the rural location of the WWTP, damaged the piping system of the bioreactors | Operational disruptions and potential damage to treatment infrastructure |
| System overloading | High inflow during peak hours (6–8 AM and 5–6 PM) led to system overloading, reducing residence time and causing pink ponds | Reduced efficiency of phycoremediation and altered pond ecology |
| Water filter maintenance | Inconsistent replacement of filters for photobioreactors, leading to chlorine in the culturing water | Adverse effects on mass culturing of algae due to chlorine in the water |
| Sludge removal | Absence of mechanical sludge removal decreased wastewater capacity in ponds, causing overloads | Overloads and reduced hydrological residence time for effective phycoremediation |
| Organic matter presence | Elevated organic matter during overloading increased turbidity, limiting light penetration | Reduced effectiveness of phycoremediation due to diminished light availability |

## 5. Discussion

Municipal or domestic wastewater, comprising discharge from households, kitchens, bathrooms, and laundry rooms, typically contains lower levels of nitrogen (N, 15–90 mgL$^{-1}$) and phosphorus (P, 5–20 mgL$^{-1}$) and has a relatively low chemical oxygen demand of less than 300 mgL$^{-1}$. This composition renders it suitable for microalgae-based treatment [33]. A major environmental issue associated with the discharge of sewage and other contaminants into water sources, especially downstream of urban areas, is eutrophication. This process exacerbates the occurrence of toxic cyanobacterial blooms, potentially producing toxins harmful to public health. In Africa, such blooms are increasingly dominating the phytoplankton communities in eutrophic lakes, posing a growing concern for water utility managers [34]. Excess nutrients, notably N and P, are key drivers of water eutrophication, a global environmental challenge [35].

The nitrogen to phosphorus (N/P) ratio in domestic wastewaters is critical for effective nutrient uptake and algal biomass growth across different algae species [36]. The ideal N/P ratio for microalgae growth ranged between 5 and 30 N:1P [37,38], with optimal ratios being strain-dependent. For instance, optimal ratios for *Chlorella* sp. and *Scenedesmus* sp. are reported to be 7 and 30, respectively [39]. Fast nitrogen uptake compared to phosphorus is consistently observed in algae growth in wastewater. Domestic wastewater with too low or too high N/P ratios can impede maximal algal growth, highlighting the importance of specific strain selection for effective wastewater treatment. In this study, the N/P ratio in Pond 6 was 2.9 and 3.4 in Pond 7 before treatment with *Chlorella* spp. The observed ratios

were too low to sustain optimal growth and treatment, yet significant nutrient reductions were still achieved. *Chlorella* spp., known for their rapid growth and adaptability to various wastewater streams, are commonly used in wastewater treatments [40]. However, for optimizing WWTPs in Africa, N/P ratios are crucial for the successful growth of *Chlorella* spp. and nutrient reduction, particularly in phosphate-sensitive river systems [41].

Only a few published case studies in Africa focused on phycoremediation in domestic wastewater treatment. These studies typically involved integrated algae pond systems (IAPSs), using naturally occurring algae in algae raceways. IAPSs employ advanced facultative ponds with in-pond anaerobic digesters followed by high-rate algae oxidation ponds with paddlewheel mixing, and utilize gravity, solar energy, and natural algae [42]. IAPSs have been implemented in South Africa, Morocco, and Zimbabwe [11]. As shown in Table 5, although optimized phycoremediation algal pond systems (OPAPSs) require longer residence times, they are less costly and require no or limited energy input, making them highly suitable for application in poor and marginalized communities.

**Table 5.** Comparison between integration algae pond systems (IAPSs) and an optimized phycoremediation algal pond systems (OPAPSs).

| Parameters | Integration of Algae Pond System (IAPSs) | Optimized Phytoremediation Algal Pond System (OPAPSs) |
| :---: | :---: | :---: |
| Infrastructure | Requires construction of raceways | Utilizes existing infrastructure of maturation ponds |
| Cost implications | Generally high financial investment | Relatively low-cost implementation |
| Energy requirement | Needs external energy input | Operates without external energy requirements |
| Residence time | Ranges from 4 to 10 days | Extends to 20 days or more |
| Mixing mechanism | Mechanical mixing using a paddle wheel | Natural mixing; potential for stratification |
| Operator expertise | Requires skilled operators | Operable by individuals without specialized skills |

## 6. Conclusions

In South Africa, as well as across the African continent, the need for renewable and efficient wastewater treatment technologies is paramount. These technologies, while technically mature, require the right support within appropriate contexts to be effective. A common obstacle to the adoption of new or improved technologies in Africa is the tendency to focus solely on the technical 'hardware', neglecting the complex interplay of social, institutional, economic, and policy factors that are crucial for success. The Motetema WWTP case study in South Africa exemplified how OPAPSs can be effectively implemented when aligned with the needs and interests of local communities, especially in collaboration with municipalities that prioritize the welfare of their constituents.

This case study highlighted the significant potential of OPAPSs in reducing risks to communities and water resources, particularly in smaller municipalities. By leveraging existing infrastructure and incorporating more effective treatment processes, such systems offer a pragmatic approach to addressing the backlog in wastewater treatment across Africa. This approach not only improves water quality and enhances downstream socio-economic activities but also contributes to the overall improvement in the quality of life by reducing eutrophication.

Furthermore, an OPAPS presents a scalable and sustainable solution for small and medium-sized communities, offering immediate and medium-term relief while also serving as a long-term sustainable option. It underscores the importance of capital investments being directed towards sustainable solutions that align with the unique challenges and opportunities in African wastewater management. The success of the Motetema WWTP

demonstrates the viability of OPAPSs in the African context, offering a roadmap for wider implementation throughout the continent.

**Supplementary Materials:** The following supporting information can be downloaded at: https://www.mdpi.com/article/10.3390/phycology4010009/s1, File S1: Culturing of algae.

**Author Contributions:** Conceptualization, P.J.O.; writing—original draft preparation, P.J.O.; writing—review and editing, Y.S. and A.-M.B.; visualization, A.-M.B.; project administration, P.J.O.; funding acquisition, P.J.O. All authors have read and agreed to the published version of the manuscript.

**Funding:** The research at Motetema WWTP was funded by The Department of Science and Technology (DST), Council of Scientific and Industrial Research (CSIR), and the Water Research Commission (WRC), South Africa.

**Institutional Review Board Statement:** Not applicable.

**Informed Consent Statement:** Not applicable.

**Data Availability Statement:** Not applicable.

**Acknowledgments:** We also want to thank the municipalities involved and providing resources for the sampling of the Motetema WWTP.

**Conflicts of Interest:** The authors declare no conflicts of interest.

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
