# Peer review of "Is Africa Ready to Use Phycoremediation to Treat Domestic Wastewater as an Alternative Natural Base Solution? A Case Study"

_phycology, doi:10.3390/phycology4010009_

Round 1
Reviewer 1 Report
Comments and Suggestions for Authors
Reviewer comments
Manuscript: Is Africa ready to use phycoremediation to treat domestic wastewater as an alternative natural base solution? A case study
ID: phycology-2867648
Date: 2024-01-31
General comments
The authors proposed an overview of microalgal wastewater treatment in South Africa. They emphasize the rationale for phycoremediation in Africa, not only with technical elements but also with social/behavioral aspects. Still, while the article is well-written and pleasant to read, the scientific content is extremely low. Therefore, it is closer to a good opinion paper than a review/research article. Consequently, it is difficult for this review to advise for or against its publication. Nevertheless, some oversights are pointed out afterward.
Major concerns
1. Is this type of article acceptable for this journal? The reviewer browsed the scope of the journal and was not able to reach a conclusion.
2. The choice of the strain genus (Chlorella) is not discussed. The same is true for the choice of a monospecific inoculation. In this view, the Scenedesmus genus is also quite efficient in phycoremediation applications. In addition, consortia usually show better results than monospecific cultures.
Minor concerns
1. Line 304, the hydraulic retention time should be specified as it is a key operating parameter of this type of system
2. Line 307, NH4+/NH3 removal should be discussed in view of pH and potential stripping.
Typos
Line 151 – remove the “s” to “days”.
Lines 222 and 224, “microalgal” should be written instead of “macroalgal”.

Author Response
In response to Reviewer 1's comments:
The authors proposed an overview of microalgal wastewater treatment in South Africa. They emphasize the rationale for phycoremediation in Africa, not only with technical elements but also with social/behavioral aspects. Still, while the article is well-written and pleasant to read, the scientific content is extremely low. Therefore, it is closer to a good opinion paper than a review/research article. Consequently, it is difficult for this review to advise for or against its publication. Nevertheless, some oversights are pointed out afterward.
Major concerns
1. Is this type of article acceptable for this journal? The reviewer browsed the scope of the journal and was not able to reach a conclusion.
We are unable to respond to this comment as we thought that the manuscript falls within the scope.
2. The choice of the strain genus (Chlorella) is not discussed. The same is true for the choice of a monospecific inoculation. In this view, the Scenedesmus genus is also quite efficient in phycoremediation applications. In addition, consortia usually show better results than monospecific cultures.
For the current study the Chlorella genus was selected since Renuka [2015] stated that Chlorella spp. is one of the most explored microalga genera in relationship with nutrient removal in different types of wastewater. Although the genus Scenedesmus was considered as part of the consortium, microscopic analyses of the pond treatment system water column before mass inoculation with Chorella spp. reveal that a very low percentage (< 6%) of the algal Scenedesmus ovaltemus did occur in the pond treatment system.The pond treatment system before mass inoculation with Chorella spp. was dominated by the algal Micractinium passillum ( 63%) followed by Eudorina elegans (22%)
REFERENCE: Renuka, N.; Sood, A.; Prasanna, R.; Ahluwalia, A.S. Phycoremediation of wastewaters: A synergistic approach using microalgae for bioremediation and biomass generation. Int. J. Environ. Sci. Technol. 2015, 12, 1443−1460. https://doi.org/10.1007/s13762-014-0700-2
Several other studies also illustrate the use of Chrorella as an alga species that shows high growth and remediation capacity. Refer to the papers of
- Kumar, A.; Ponmani, S.; Sharma, G. K.; Sangavi, P.; Chaturvedi, A. K.; Singh, A.;Malyan, S. K.; Kumar, A;, Khan, S. A.; Shabnam, A. A.; Jigyasu, D. K.; Gull, A. Plummeting toxic contaminates from water through phycoremediation: Mechanism, influencing factors and future outlook to enhance the capacity of living and non-living algae. Res., 2023, 239, 117381. https://doi.org/10.1016/j.envres.2023.117381
- Koul, B.; Sharma, K.; Shah, M. P. Phycoremediation: A sustainable alternative in wastewater treatment (WWT) regime. Tech. Innovat., 2022, 25, 102040. https://doi.org/10.1016/j.eti.2021.102040
Minor concerns
1. Line 304, the hydraulic retention time should be specified as it is a key operating parameter of this type of system
Residence time for the Motetema pond treatment systems were between 26 and 29 days depending on environmental conditions, for example rain conditions.
2. Line 307, NH4+/NH3 removal should be discussed in view of pH and potential stripping.
The total nitrogen mg l-1 was removed by 73% after mass inoculation with Chorella spp. while ammonia was reduced by 99.4%. The pH of the pond system increases from an average of 8.2 to 9.1 after mass inoculation with Chorella spp. It was evident that the reductions were strongly related to the increase in pH. According to the literature (EPA, 2011), the rate of gaseous NH3 losses to the atmosphere is primarily a function of a high pH, surface to volume ratio of the maturation pond, temperature, and the mixing conditions in the maturation pond. A high alkaline water pH shifts the equilibrium of NH3 gas and NH4 towards gaseous NH3 production, while the mixing conditions in the maturation pond affect the magnitude of the mass transfer coefficient.
REFERENCE: Environmental Protection Agency (EPA) Principles of Design and Operations of Wastewater Treatment Pond Systems for Plant Operators, Engineers, and Managers. EPA/600/R-11/088 | August 2011 | www.epa.gov /nrmrl
To address this query we included the following:
Line 360-365: "According to the literature [32] the rate of gaseous NH3 losses to the atmosphere is primarily a function of a high pH, surface to volume ratio of the maturation pond, temperature, and the mixing conditions in the maturation pond. A high alkaline water pH shifts the equilibrium of NH3 gas and NH4 towards gaseous NH3 production, while the mixing conditions in the maturation pond affect the magnitude of the mass transfer coefficient."
Typos
Line 151 – remove the “s” to “days”.
To address this comment, the "s" was removed and now reads "day"
Lines 222 and 224, “microalgal” should be written instead of “macroalgal”.
To address this comment, "macroalgal" was replaced with "microaglal" in both imnstances.
Reviewer 2 Report
Comments and Suggestions for Authors
Dear authors,
I carefully read your manuscript titled "Is Africa ready to use phycoremediation to treat domestic wastewater as an alternative natural base solution? A case study". I recommend for your manuscript a minor revisions.
All my suggestions are in the attached PDF.
Your article provides valuable insights into the potential of phycoremediation as a natural and cost-effective solution for domestic wastewater treatment in Africa. It addresses the urgent need to improve sanitation access, a critical aspect in achieving the Sustainable Development Goals.
The case study on South Africa illuminates the specific challenges faced by the continent in wastewater management, highlighting issues such as limited treatment plant capacity, fluctuating pollution levels due to unregulated discharges, and high operational costs. Your analysis underscores the importance of addressing these challenges to ensure sustainable water and sanitation infrastructure across the continent.
While the article is well-written overall, some improvements are necessary, particularly in the citation formatting as suggested. Additionally, it would be beneficial to provide further empirical evidence and practical experiences of phycoremediation implementation in similar contexts.
In conclusion, your manuscript represents a significant contribution to the literature on wastewater treatment in Africa.
Regards

Author Response
In response. to Reviewer 2's comments the following changes were made:
Lines 29-34 were moved to Line 116 and the focus of the review re-defined to be more on microalgal sp. and now reads as follows:
"In response, this paper highlights the optimization of phycoremediation, a process offering multiple benefits. Phycoremediation is a low-cost, electricity-free, and chemical-free approach that utilizes existing infrastructure. It is environmentally friendly and straightforward to implement and operate, making it particularly suitable for enhancing treatment capacity in small, rural wastewater treatment plants (WWTPs). This review will mostly focus on the direct use of microagal species due to their high potential to convert solar to chemical energy [12 and references within; 13], and posits that optimizing phycoremediation presents a viable, eco-sensitive, and cost-effective medium-term solution for these areas. It emphasizes the feasibility of this method in terms of resource utilization and operational simplicity, potentially revolutionizing wastewater treatment in rural Africa."
Lines 130-134: "phytoremediation"' were redefined and additional references added as recommended.
Line 217: colour was changed from red to black
Line 220: a space was inserted between the end of the sentence and the beginning of the following sentence.
Line 377: The heading was changed to "Discussion" as suggested.
New references were added and include the following:
- Kumar, A.; Ponmani, S.; Sharma, G. K.; Sangavi, P.; Chaturvedi, A. K.; Singh, A.;Malyan, S. K.; Kumar, A;, Khan, S. A.; Shabnam, A. A.; Jigyasu, D. K.; Gull, A. Plummeting toxic contaminates from water through phycoremediation: Mechanism, influencing factors and future outlook to enhance the capacity of living and non-living algae. Res., 2023, 239, 117381. https://doi.org/10.1016/j.envres.2023.117381
- Salah, A.; Sany, H.; El-Sayed, A. E.-K. B.; El-Bahbohy, R. M.; Mohamed, H. I.; Amin, A. Growth Performance and Biochemical Composition of Desmodesmus sp. Green Alga Grown on Agricultural Industries Waste (Cheese Whey). Water Air Soil Poll., 2023, 234(12), 770. https://doi.org/10.1007/s11270-023-06780-0
- Koul, B.; Sharma, K.; Shah, M. P. Phycoremediation: A sustainable alternative in wastewater treatment (WWT) regime. Tech. Innovat., 2022, 25, 102040. https://doi.org/10.1016/j.eti.2021.102040